# *Humulus lupulus* L. Extract Protects against Senior Osteoporosis through Inhibiting Amyloid *β* Deposition and Oxidative Stress in APP/PS1 Mutated Transgenic Mice and Osteoblasts

**DOI:** 10.3390/molecules28020583

**Published:** 2023-01-06

**Authors:** Tianshuang Xia, Jiabao Zhang, Yunxiang Guo, Yiping Jiang, Fangliang Qiao, Kun Li, Nani Wang, Ting Han, Hailiang Xin

**Affiliations:** 1School of Pharmacy, Navy Medical University, Shanghai 200433, China; 2Department of Medicine, Zhejiang Academy of Traditional Chinese Medicine, Hangzhou 310007, China

**Keywords:** amyloid *β*, *Humulus lupulus* L., senior osteoporosis, APP/PS1 mice, oxidative stress

## Abstract

As aging progresses, *β*-amyloid (A*β*) deposition and the resulting oxidative damage are key causes of aging diseases such as senior osteoporosis (SOP). *Humulus lupulus* L. (hops) is an important medicinal plant widely used in the food, beverage and pharmaceutical industries due to its strong antioxidant ability. In this study, APP/PS1 mutated transgenic mice and A*β*-injured osteoblasts were used to evaluate the protective effects of hops extracts (HLE) on SOP. Mice learning and memory levels were assessed by the Morris water maze. Mice femurs were prepared for bone micro-structures and immunohistochemistry experiments. The deposition of A*β* in the hippocampus, cortex and femurs were determined by Congo red staining. Moreover, protein expressions related to antioxidant pathways were evaluated by Western blotting. It was found that HLE markedly improved learning abilities and ameliorated memory impairment of APP/PS1 mice, as well as regulated antioxidant enzymes and bone metabolism proteins in mice serum. Micro-CT tests indicated that HLE enhanced BMD and improved micro-architectural parameters of mice femur. More importantly, it was discovered that HLE significantly reduced A*β* deposition both in the brain and femur. Further in vitro results showed HLE increased the bone mineralization nodule and reduced the ROS level of A*β*-injured osteoblasts. Additionally, HLE increased the expression of antioxidant related proteins Nrf2, HO-1, NQO1, FoxO1 and SOD-2. These results indicated that *Humulus lupulus* L. extract could protect against senior osteoporosis through inhibiting A*β* deposition and oxidative stress, which provides a reference for the clinical application of hops in the prevention and treatment of SOP.

## 1. Introduction

Senior osteoporosis (SOP) is a type of metabolic disease characterized by osteopenia, bone micro-structure degeneration and fracture. Aging is a major pathogenic factor causing SOP. Along with aging, the body will induce oxidative stress through releasing excessive reactive oxygen species (ROS), further decreasing bone formation in osteoblasts and increasing bone resorption in osteoclasts, eventually leading to a bone homeostasis imbalance [1]. According to statistics, more than one-fifth of men and one-third of women in the world suffer from osteoporosis at the age of 50 or above [2]. Furthermore, it is estimated that the number of people suffering from osteoporosis in China will exceed 200 million by 2050 [3], which is a crucial public health problem to be solved. In recent years, there is increasing evidence suggesting that SOP patients are more likely to have memory impairment, even Alzheimer’s disease (AD) [4], which is associated with oxidative stress and *β*-amyloid (A*β*) protein plaque deposition. However, these are all relatively unclear pathogeneses of SOP. Current clinical drug treatments for SOP mainly include estrogen therapy, bisphosphonates supplementation, as well as calcium and active vitamin D, which cause more side effects and lack a clear target. Therefore, there is a desperate need to elucidate the pathogenesis of SOP and to find appropriate alternative drugs for SOP with few adverse effects.

*Humulus lupulus* L. (hops) is an important medicinal plant widely used in the food, beverage and pharmaceutical industries due to its strong antioxidant capacity [5], and has long medicinal history in China for digestive diseases, tuberculosis, insomnia and forgetfulness. In Europe, hops is used for hot flushes during the menopause and postmenopausal osteoporosis [6]. In addition, hops has been proven to have a strong antioxidation effect and is a potential antioxidant [7,8]. It has been reported that phenolic acids (including hydroxybenzoic acids, hydroxycinnamic acids and hydroxyphenylpropanoic acids) and flavonoids (mainly anthocyanins, flavones, flavonols and isoflavonoids) are the main effective components of hops, which contribute to its antioxidant capacity. Our previous studies have also discovered that hops extract could prevent ovariectomy-induced osteoporosis in mice and regulate the activities of osteoblasts and osteoclasts through attenuating oxidative stress [9,10,11]. In addition, xanthohumol, a unique isoflavone in hops, has been found to have a potent effect on A*β*-induced oxidative damage and bone loss in APP/PS1 mice and osteoblasts [12,13]. These studies demonstrated that hops have potential as an antioxidant with anti-A*β* deposition and anti-SOP properties.

A*β* aggregation and deposition in the cerebrum is a significant pathological feature in AD patients. A*β* deposition can cause neurotoxicity and oxidative stress, which leads to wide neurodegeneration [14]. More researchers have found that A*β* not only exists in the cerebrum, but also in the bone [15], and A*β*_42_ is often abnormally elevated in osteoporosis patients [16]. In addition, progressively more clinical results have demonstrated that most senile dementia patients suffer from bone diseases and have a high risk of fractures [4]. SOP may be caused by A*β* deposition in femurs, which affect the activities of osteoblasts and osteoclasts [17]. More importantly, it has been discovered that in APP/PS1 mutated transgenic mice, both brain and bone tissues showed A*β* deposition accompanied with peroxidation injury, and antioxidants could improve the cognitive ability and bone loss caused by this A*β* deposition and oxidative damage [18]. It is therefore speculated that antioxidants may prevent bone loss caused by A*β* deposition and oxidative damage.

In view of the strong antioxidant effect of hops, the present study employed APP/PS1 mice to investigate the effect of hops on memory deficit and bone loss induced by A*β* deposition, and probed its potential mechanism using A*β*-injured osteoblasts, which can provide more references for the prevention and treatment of SOP.

## 2. Results

### 2.1. HLE Prevented Spatial Memory Deficit of APP/PS1 Mice

The Morris water maze (MWM) task was conducted to estimate whether hops extract could improve the long-term spatial memory of APP/PS1 mice. After 2 months administration, 11-month-old mice were prepared for the MWM task. The MWM task included two parts: task acquisition (days 1–5) and probe trial (day 6). During the experiment, the tank was videotaped from above, and all the relevant data were recorded. Detailed experimental steps are shown in “Section 4.3”. As shown in Figure 1A, during 5-day training, the escape latency of mice to find and load upon the platform decreased progressively. From the third training day to the last training day, the latency of APP/PS1 mice was significantly longer than that of wild-type mice (*p* < 0.001), and the mice in treatment groups had a lower latency to load upon the platform compared with that of APP/PS1 mice during the last two training days (*p* < 0.05). On the probe test day, the swimming time spent in the target zone is shown in Figure 1B. We observed significant differences in swimming time between APP/PS1 mice and mice in treatment groups (*p* < 0.05). APP/PS1 mice swam for less time in the target zone than the wild-type mice, and the mice treated with HLE or N-acetyl-L-cysteine (NAC, a kind of antioxidant, used as positive control in this study) spent an increased amount of time in the target zone. Moreover, as shown in Figure 1C, it was observed that the number of platform crossings in APP/PS1 mice was significantly lower than that in the wild-type mice (*p* < 0.001), while treatment with HLE (2 g/kg) or NAC significantly increased the number of platform crossings (*p* < 0.05). The swimming tracking showed that APP/PS1 mice mostly swam in the quadrant far from the target quadrant, while mice in wild-type group and other treatment groups mostly swam near the platform (Figure 1E). However, the treatment did not affect the swimming speed compared with APP/PS1 mice, as shown in Figure 1D (*p* > 0.05). These results suggested that HLE had a good effect on improving the learning and memory abilities and could prevent spatial memory deficit of APP/PS1 mice.

### 2.2. HLE Reduced Aβ Deposition Both in the Brain and Bone of APP/PS1 Mice

To determine the effect of HLE on A*β* deposition, we applied Congo red strain to observe the pathological changes in A*β* plaque in the brain and femur. The brownish red or orange precipitates were positive for A*β* (Figure 2A). In the brain, A*β* plaque in mice hippocampi and cortexes was measured, and it was found that the amount of A*β* plaque in APP/PS1 group mice was significantly more than that of the wild-type group (*p* < 0.001). HLE or NAC markedly decreased the amount of A*β* plaque in hippocampi and cortexes of APP/PS1 mice (*p* < 0.001). Moreover, we observed more A*β* deposition in the femurs of APP/PS1 mice than in wild-type mice (*p* < 0.01), while HLE or NAC could ameliorate the accumulation of A*β* in femurs of APP/PS1 mice (*p* < 0.01) (Figure 2B–D). These results suggested that HLE could reduce A*β* deposition both in the brains and bones of APP/PS1 mice.

### 2.3. HLE Improved Bone Mineral Density (BMD) and Bone Microarchitecture of APP/PS1 Mice

Micro-CT was conducted to observe the bone structural properties of trabecular bone and BMD in femurs. As shown in the micro-CT images (Figure 3A), there was larger gap space in the ROI region in APP/PS1 mice compared to mice in wild and treatment groups. As shown in Figure 3B, femur BMD in APP/PS1 mice was significantly decreased (*p* < 0.01), while HLE or NAC could reverse this decrease and enhance the BMD (*p* < 0.05). The bone volume fraction (BVF) indicates the ratio of bone volume to total volume. Figure 3C showed that the BVF of APP/PS1 mice was significantly decreased compared with that of wild-type mice (*p* < 0.01), while HLE or NAC could markedly increase the femur BVF in APP/PS1 mice (*p* < 0.05). In addition, as shown in Figure 3D–F, the morphologic parameters of trabecular number (Tb.N.) and trabecular thickness (Tb.Th.) decreased, while trabecular separation (Tb.Sp.) increased significantly in APP/PS1 mice when compared with those in wild group. On the contrary, HLE or NAC markedly reversed these changes in trabecular morphological parameters by increasing the Tb.N. and Tb.Th. and decreasing the Tb.Sp., indicating that HLE had an excellent bone protection effect.

### 2.4. HLE Relieved Oxidative Stress and Regulated Bone Metabolism in APP/PS1 Mice

As shown in Figure 4A, the superoxide dismutase (SOD) activity in the serum of APP/PS1 mice decreased to 4.341 ± 0.45 U/mL, significantly less than that of the wild-type mice (7.663 ± 0.34 U/mL) (*p* < 0.001). Moreover, osteocalcin (OCN), which plays an important role in regulating bone metabolism, was activated less in APP/PS1 mice (4.237 ± 0.41 ng/mL) than that in wild-type mice (11.250 ± 0.43 ng/mL) (Figure 4B, *p* < 0.001). HLE or NAC could increase the SOD and OCN levels of APP/PS1 mice, respectively (*p* < 0.001), and there were no significant differences between the two dose groups. Inflammatory cytokines interleukin-1*β* (IL-1*β*) and interleukin-6 (IL-6) levels in the serum of APP/PS1 mice significantly increased from 5.028 ± 0.46 ng/L and 35.51 ± 1.18 ng/L in wild-type mice to 19.42 ± 0.68 ng/L and 51.70 ± 2.27 ng/L (*p* < 0.001), respectively. After treatment, HLE or NAC reversed the high level of IL-1*β* and IL-6 in the serum of APP/PS1 mice to almost ordinary levels (Figure 4C,D, *p* < 0.001).

The bone metabolism index osteoprotegerin (OPG) and oxidative stress indexes nuclear factor erythroid 2-related factor 2 (Nrf2), forkhead box O1 (FoxO1) and SOD-2 in mice femurs were measured by immunohistochemistry. As shown in Figure 5A, compared with wild-type mice, the expression of OPG in femurs of APP/PS1 mice was significantly reduced (*p* < 0.01). High doses of HLE significantly improved this inhibition (*p* < 0.01), indicating that endogenous A*β* reduced OPG content, and HLE might play a bone protective role by promoting the expression of OPG. As shown in Figure 5B–D, Nrf2, FoxO1 and SOD-2 expression in femurs of APP/PS1 mice was significantly reduced compared with that in wild-type mice (*p* < 0.01), while high doses of HLE could return it to a normal level (*p* < 0.01), suggesting that HLE might alleviate oxidative stress through regulating Nrf2 and FoxO1 pathways.

### 2.5. HLE Improved Cell Activities and Alleviated Oxidative Stress in Aβ-Injured Osteoblasts

To further verify the effect of hops on alleviating oxidative stress and promoting bone formation through inhibiting A*β* deposition, osteoblasts were injured by A*β* for in vitro study. Bone mineralization levels were measured by Alizarin red staining. As shown in Figure 6A, all doses of HLE significantly increased the bone mineralization nodule in A*β*-injured osteoblasts, proving that hops could promote bone mineralization and bone formation. As shown in Figure 6B, A*β* significantly improved the ROS release in osteoblasts, while HLE markedly reduced the ROS level in a dose-dependent manner (*p* < 0.01). In addition, oxidative stress related Nrf2 and FoxO1 pathways were measured by Western blotting. As shown in Figure 6C–E, A*β* markedly reduced the expression of Nrf2, heme oxygenase-1 (HO-1), NAD(P)H:quinone oxidoreductase 1 (NQO1), FoxO1 and SOD-2 compared to the control group. After treatment, HLE or NAC significantly reversed the decreased expression of Nrf2, HO-1, NQO1, FoxO1 and SOD-2 in A*β*-injured osteoblasts, indicating that HLE could alleviate oxidative stress caused by A*β* deposition through activating Nrf2 and FoxO1 pathways.

## 3. Discussion

Osteoporosis is a degenerative chronic disease closely related to aging. It is crucial to fully recognize the potential risks and benefits of diagnosing and treating osteoporosis in elderly people. The high consumption of beer all over the world leads us to focus on hops plants that are essential in beer brewing, and our previous studies have detailed the potent effects of hops on relieving oxidative stress and alleviating bone loss. However, details of the mechanism of hops on preventing senior osteoporosis have not been clarified. In this study, it was determined for the first time that hops improved the learning ability and alleviated bone loss of APP/PS1 mice through inhibiting A*β* deposition in both the brain and bone. In vivo and in vitro experiments revealed hops could relieve A*β*-induced oxidative stress and protect against SOP.

In agreement with the clinical and epidemiological evidence, it has been found that A*β* deposition and the dysfunction of antioxidant system play an important role in the pathogenesis of SOP and AD [19]. Extracellular amyloid plaques and intracellular neurofibrillary tangles in the brain are primary indicators of AD. It has been demonstrated that the mutations in the APP gene, presenilin (PS) 1, potentially lead to early onset of AD [20]. In mice, the APP transgene combined with a PS1 transgene yielded A*β* plaques with earlier onset than the single transgenic sample [21]. A*β* directly or indirectly modulates mitochondrial function and induces oxidative stress, which in turn enhances A*β* synthesis and aggregation. In this study, APP/PS1 mutated transgenic mice were used to mimic the SOP condition, and we found that there was A*β* deposition not only in the brain but also in the femurs of APP/PS1 mice, which was an important trigger of memory impairment and osteoporosis. After treatment, it was discovered that hops could both alleviate memory impairment and attenuate bone loss. More importantly, the A*β* amount in mice hippocampi, cortexes and femurs were markedly decreased after treatment with HLE, proving that hops could alleviate senior osteoporosis and dementia in APP/PS1 mice through removing A*β* deposition.

The histomorphometric parameters of the trabecular bone obtained from micro-CT analysis can predict osteopenia and deterioration of bone quality, and BMD is known as the golden indicator for diagnosing osteoporosis [22]. The present study found that the femur BMD of APP/PS1 mice decreased significantly and the bone micro-structure was severely damaged, indicating that A*β* deposition led to bone loss. Hops could improve the micro-architecture, enhance the BMD, and increase the trabecular parameters in the femurs of APP/PS1 mice, suggesting that hops was effective in both preserving bone mass and rescuing the deterioration of bone micro-architecture when damaged by A*β* deposition. During bone remodeling, osteoblastic bone formation and osteoclastic bone resorption are a synergistic action. OCN and OPG are important enzymes characterized by the capacity of osteoblastic bone formation [23], while the concentrations of IL-1*β* and IL-6 in serum were associated with increased ROS and osteoclastic bone resorption [24]. IL-1*β* and IL-6 not only directly stimulate osteoclastogenesis and bone resorption but also stimulate RANKL production in osteoblastic cells in a synergistic fashion. In this study, hops can significantly increase the OCN and OPG expression and inhibit the serum IL-1*β* and IL-6 levels in APP/PS1 mice, indicating the potential of hops in maintaining bone homeostasis.

A*β* deposition could interfere with the mitochondria, which causes cells to stop breathing and subsequently oxidative stress [25,26]. Studies have increasingly shown that excessive oxidative stress can lead to memory impairment and induce osteoporosis. On one hand, excess ROS damages osteogenic activity and osteoblast dysfunction, eventually resulting in osteoporosis [27]. On the other hand, dementia is mainly caused by the accumulation of A*β* protein plaque and overexpression of hyperphosphorylated tau protein in neurons [28], all of which ultimately induce neurotoxic effects via the upregulation of ROS in the brain. Thus, oxidative stress is seen as a risk factor for both AD and SOP. In this process, SOD is the major enzymatic scavenger in the antioxidant-defense system. The transgenic mice with overexpression of APP mutant and deficiency of Mn-SOD had elevated oxidative stress and significantly increased levels of brain A*β* plaque. Conversely, Mn-SOD was overexpressed in APP/PS1 mice and they exhibited an increased antioxidant defense capability in the brain and a reduced level of A*β* plaque [29]. In our study, we observed significantly lower SOD activities in both the serum and femurs of APP/PS1 mice, and higher ROS levels in A*β*-injured osteoblasts. Fortunately, hops could increase the SOD level and reduce active oxygen release, thus relieving oxidative stress.

As a key redox-sensitive transcription factor, Nrf2 is conducive to maintaining cellular redox homeostasis and improving oxidative injury. Expressions of antioxidant enzymes SOD and CAT are intimately related with Nrf2 signaling activation [30]. The by-products catalyzed by HO-1 have potent ROS scavenging activity, while NQO1-induced by-products prevent DNA oxidative damage caused by environmental stress agents [31]. FoxO1, the major member in the Forkhead box O family, counteracts ROS generation by upregulating antioxidant enzymes. FoxO can also affect the proliferation and differentiation of osteoblasts through its regulation of the redox balance [32]. In this study, it was discovered that hops could increase the Nrf2, FoxO1 and SOD-2 expression in the femurs of APP/PS1 mice and promote the expression of Nrf2 and FoxO1 pathway-related proteins in A*β*-injured osteoblasts, suggesting that hops might alleviate oxidative stress and SOP through regulating Nrf2 and FoxO1 pathways.

Collectively, we have, for the first time, demonstrated that hops extract protects against A*β*-induced senior osteoporosis primarily by ameliorating memory impairment, enhancing BMD and trabecular bone structure and improving osteoblastic MC3T3-E1 cell activities in APP/PS1 mutated transgenic mice. As for its mechanism, removing A*β* deposition in the brain and bone and its inducing of oxidative stress plays a crucial role.

## 4. Materials and Methods

### 4.1. Reagents

*Humulus lupulus* L. was obtained from the Anguo Traditional Chinese medicine market (Hebei, China) and identified through microscope identification and HPLC to test its quality (Figure 7). An amount of 150 g *Humulus lupulus* L. was extracted by 2.25 L 75% ethanol at 80 °C for 2 h twice. The filtrate was concentrated at 40 °C under reduced pressure. The final extract was standardized by ICE-3 and xanthohumol, a special component in hops, to evaluate the quality by HPLC. The content of xanthohumol in the extract was 0.55%. The final extract was stored at −20 °C.

NAC and SOD assay kits were purchased from Shanghai Biyotime Biotechnology Co., Ltd. Enzyme-linked immunosorbent assay (ELISA) kits for determination of OCN, IL-1*β* and IL-6 were purchased from Nanjing Jiancheng Bioengineering Institute. Antibodies against Nrf2, HO-1, NQO1, FoxO1 and SOD-2 were purchased from Abcam. All materials were dissolved in 0.5% CMC-Na solution.

### 4.2. Animals and Treatment

Male APP/PS1 mutated transgenic mice (9 months old, 25–30 g) were obtained from the Nanjing Biomedical Research Institute of Nanjing University (Certificate No. SCXK 2015-0001, Nanjing, China), and housed four per cage, maintained under constant temperature (23 ± 1 °C) and humidity (60 ± 10%) under a 12 h light/dark cycle (light from 7:30 am to 7:30 pm). Mice were freely provided with water and food and divided equally into five groups of ten mice, namely CON, APP/PS1, APP/PS1 + NAC (100 mg/kg/d), APP/PS1 + HLE-L (1 g/kg/d) and APP/PS1 + HLE-H (2 g/kg/d). The selection of the drug doses in the experiment was based on our previous study [9]. The experimental dosage was adjusted according to the weight by 0.1 mL/g, and all drugs were given 6 days a week for 2 months. The wild mice were treated with the same volume of 0.5% CMC-Na solution by intragastric administration. In the end, mice were fasted for 12 h, followed by anesthetization by injection of 3 mL/kg 10% (*w*/*v*) chloral hydrate. Serums were centrifuged and stored at −80 °C for biochemical assay, and femurs were prepared for micro-CT and immunohistochemistry experiments. All studies were conducted in accordance with the NIH publication and approved by the Committee on Ethics of Medical Research Second Military Medical University.

### 4.3. Morris Water Maze Task

The Morris water maze task was employed to test spatial learning and memory ability of mice in this study [33]. The task consisted of a circular pool (180 cm in diameter and 45 cm in height) and a transparent escape platform (9 cm in diameter and 29 cm in height). The platform was submerged 1 cm below the surface of the water and placed in one quadrant named the target quadrant. Four visual cues were placed around the tank (one triangle, one square, one circle and one hexagon) and the water in the pool was maintained at 24 ± 1 °C, and contained titanium white powder. On the first day, all test mice were habituated to swimming for 60 s freely in the tank without the escape platform. During the next five days, the mice were trained to swim to the platform placed in the center of the target quadrant within 60 s, with three trials per day in a section, and the interval between the two trails was 30 min. The time (latency) spent searching for platform was recorded. On the probe test day, the animals were admitted to freely swim in the tank without the platform for 60 s, and the time spent in the target quadrant, the number of crossings of the platform and the swimming velocity were recorded.

### 4.4. Congo Red Staining

After mice were sacrificed, the whole brain and right femurs were picked and fixed in 4% paraformaldehyde, then placed in 30% sucrose solution until they sunk to the bottom. The tissue was continuously cut, and every slice (10 μm, three sections for one mouse) was stained with Congo red to determine the total amount of A*β* plaque, as described previously [13]. In the end, the hippocampus, cortex and femur slices were mounted on slides for detection using an Olympus microscope with DP-70 software.

### 4.5. Micro-CT and Biochemical Marker Measurement

Mice femur micro-architecture was measured with a micro-CT scanner (GE eXplore Locus SP). BMD and trabecular bone parameters of BVF, Tb.N., Tb.Th. and Tb.Sp. were auto calculated by the computer.

Mice serum was centrifuged at 3000 r/min for 10 min, and the supernatant was collected for the biochemical markers assay. The SOD, OCN, IL-1*β* and IL-6 levels were measured by commercially available assay kits in accordance with the manufacturer’s instructions.

### 4.6. Immunohistochemistry

The femur section was separated for immunohistochemistry by xylene and hydrated in ethanol of graded concentrations. The cross-section was bathed in sodium citrate buffer (pH = 6.0) and heated. Then, the section was incubated in 3% hydrogen peroxide for 25 min and 3% BSA solution for 30 min, successively. After that, the sections were incubated with primary and secondary antibodies at 4 °C, successively. Finally, slides were incubated by a DAB staining kit and stained with hematoxylin-eosin, and the positive section was stained brown-yellow. The coverage of positive staining was calculated with Image-pro plus software.

### 4.7. Cell Cultures and Treatment

Osteoblastic MC3T3-E1 cells were purchased from the typical Culture Committee Cell Library of the Chinese Academy of Sciences, Shanghai, China, and cultured in DMEM containing 10% FBS in a humidified atmosphere of 5% CO_2_ at 37 °C. When reaching 80% confluence, osteoblasts could be used for follow-up studies. According to the experiment design, osteoblasts were incubated overnight and then pretreated with different concentration of HLE (4, 20 and 100 μg/mL) or NAC (2.5 mM) for 4 h. After that, the plate was removed from the incubator and 1 μL of 1 mM A*β*_1–42_ oligomer mother liquor was added to each well, resulting in a final concentration of 10 μM A*β*. Osteoblasts were incubated for another 44 h, and cell growth in the orifice without the treatment was used as the control group.

### 4.8. Osteoblastic Mineralization and ROS Analysis

The cultured osteoblasts were added to 24-well plates overnight (5 × 10^4^ cells/well) and then cultured with osteogenic differentiation medium (10 nM dexamethasone, 50 μg/mL ascorbic acid and 10 mM *β*-glycerophosphate) for 18 days. Then, cells were cultured in HLE or NAC, which contained 10 μM A*β*, for another 2 days. Osteoblasts were fixed in ice-cold 4% paraformaldehyde for 10 min, and then dyed with 0.1% Alizarin red solution at 37 °C for 30 min. After washing, osteoblasts were completely dissolved in 10% cetylpyridinium chloride for 15 min, and then measured at 570 nm.

For intracellular ROS measurements, osteoblasts were treated with reagents for 48 h and then incubated with 5 μM DCFHDA for 30 min at room temperature. Finally, the intracellular ROS level was analyzed with a flow cytometer according to the instructions of the ROS Assay Kit.

### 4.9. Western Blotting

Osteoblasts were seeded in 6-well plates with a density of 1 × 10^5^ cells/mL. After 48 h treatment, the osteoblasts were lysed and centrifuged at 12,000 r/min for 10 min. The cell lysis solution was separated onto 10% sodium dodecyl sulfate-polyacrylamide gels and electrically blotted onto a polyvinylidene fluoride membrane. Membranes were blocked with 5% BSA for 1 h and then incubated with primary antibodies overnight at 4 °C. These targets were immunoblotted in the same membrane from which GAPDH was immunoblotted. After that, TBST was used to wash membranes three times, and the membranes were incubated with horseradish peroxidase-conjugated goat anti-rabbit secondary antibodies at 37 °C for 1 h. Membranes were visualized by enhanced chemiluminescent (ECL) reagents and imaged using the Gel imaging system. For protein bands with similar KD values, we used the removal solution to remove the previous protein imprint, and then carried out another incubation for other antibodies.

### 4.10. Statistical Analysis

All data were expressed as mean ± standard error of mean (SEM). Statistical significance was set at *p* < 0.05 and determined by one-way analysis of variance and the Student–Newman–Keuls test for multiple comparisons. GraphPad Prism (version 5.0) was used for statistical analysis.

## Figures and Tables

**Figure 1 molecules-28-00583-f001:**
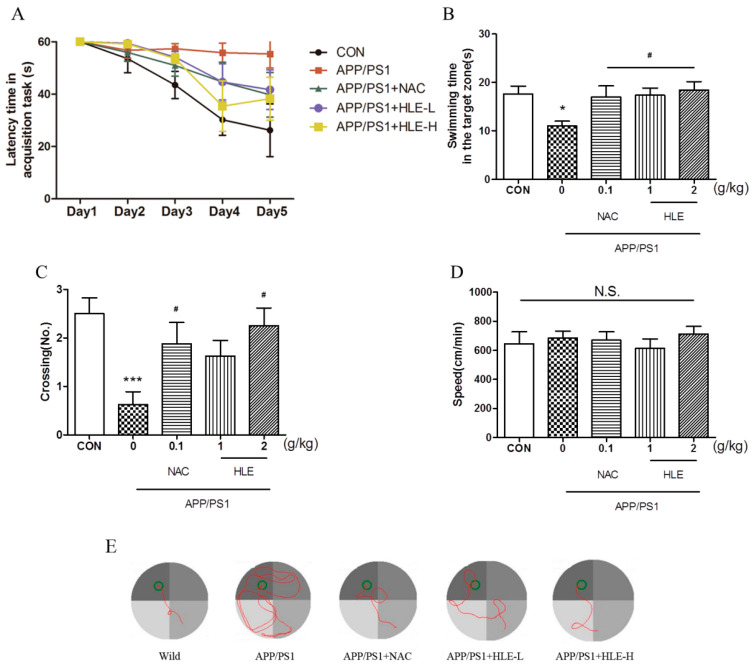
Effects of HLE on spatial memory impairment of APP/PS1 transgenic mice in the Morris water maze task. (**A**) Escape latency time of tested mice over 5 days of MWM task acquisition; (**B**) the swimming time; (**C**) the crossing numbers; (**D**) swimming speed in the target zone on the probe test day and (**E**) path taken by one random rat in each group during MWM probe trial (*n* = 8). (* *p* < 0.05, *** *p* < 0.001, compared with the control (CON) group; # *p* < 0.05, compared with the APP/PS1 group.)

**Figure 2 molecules-28-00583-f002:**
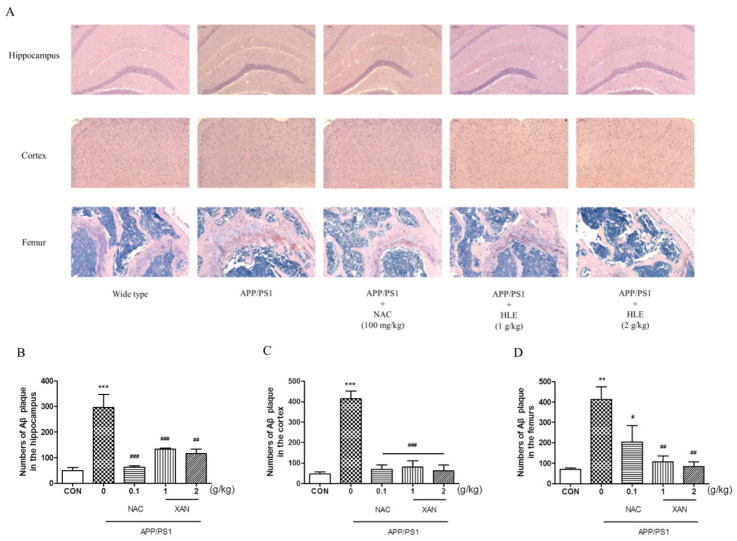
The effects of HLE on A*β* plaque in the hippocampus, cortex and femurs. (**A**) Congo red-positive plaque in the hippocampus, cortex or femurs of APP/PS1 mice; (**B**–**D**) quantitative analysis of amount of A*β* (*n* = 3). (** *p* < 0.01, *** *p* < 0.001, compared with the CON group; # *p* < 0.05, ## *p* < 0.01, ### *p* < 0.001 compared with the APP/PS1 group.)

**Figure 3 molecules-28-00583-f003:**
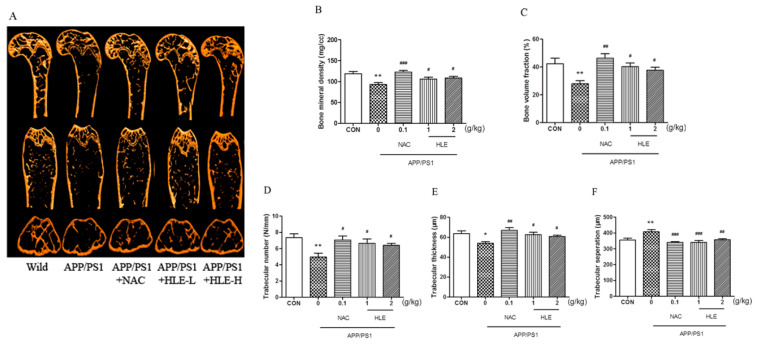
Effects of HLE on bone mineral density and the structures of trabecular bone in femurs of APP/PS1 mice. (**A**) Micro-CT images of the ROI region in the longitudinal section, transverse section and 3-D architecture; (**B**–**F**) trabecular bone parameter analysis of (**B**) BMD; (**C**) BVF; (**D**) Tb.N.; (**E**) Tb.Th.; and (**F**) Tb.Sp. in the distal femur region in APP/PS1 mice (*n* = 8). (* *p* < 0.05, ** *p* < 0.01, compared with the CON group; # *p* < 0.05, ## *p* < 0.01, ### *p* < 0.001 compared with the APP/PS1 group.)

**Figure 4 molecules-28-00583-f004:**
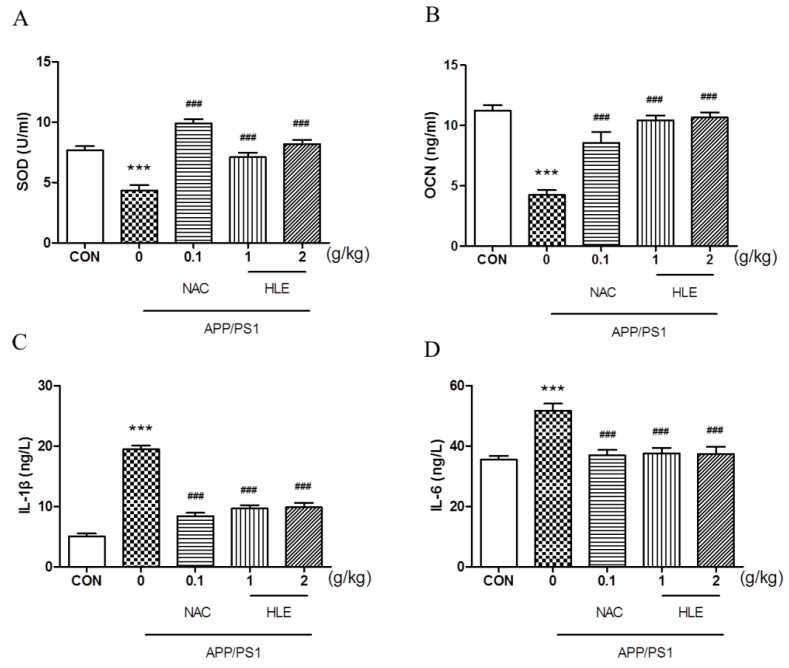
Effects of HLE on (**A**) SOD; (**B**) OCN; (**C**) IL-1*β*; and (**D**) IL-6 in serum of APP/PS1 mice by the Elisa test (*n* = 8). (*** *p* < 0.001, compared with the CON group; ### *p* < 0.001 compared with the APP/PS1 group).

**Figure 5 molecules-28-00583-f005:**
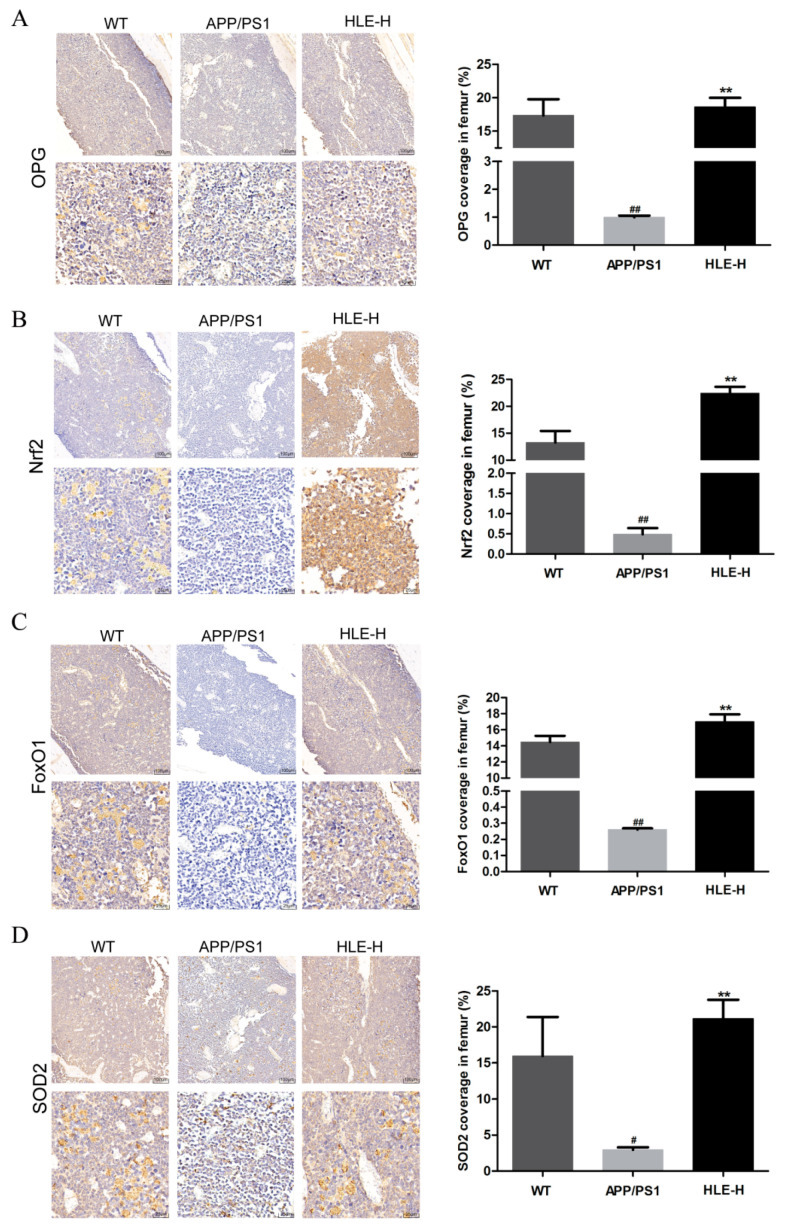
Effects of HLE on (**A**) OPG; (**B**) Nrf2; (**C**) FoxO1; and (**D**) SOD2 expression in femurs of APP/PS1 mice by immunohistochemistry (*n* = 3). (# *p* < 0.05, ## *p* < 0.01, compared with the WT group; ** *p* < 0.01 compared with the APP/PS1 group).

**Figure 6 molecules-28-00583-f006:**
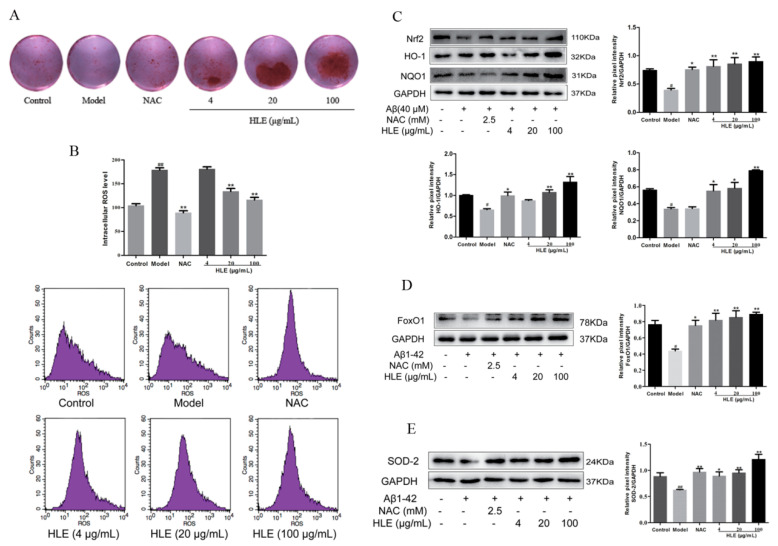
Effects of HLE on bone formation and oxidative stress in A*β*-injured osteoblastic MC3T3-E1 cells. (**A**) The bone mineralization nodule determined by Alizarin red staining; (**B**) intracellular ROS levels determined by flow cytometry; (**C**–**E**) relative expression of Nrf2, HO-1, NQO1, FxoO1 and SOD-2 determined by Western blotting (*n* = 3). (# *p* < 0.05, ## *p* < 0.01, compared with the CON group; * *p* < 0.05, ** *p* < 0.01 compared with the model group. Model group means osteoblasts only treated with A*β*).

**Figure 7 molecules-28-00583-f007:**
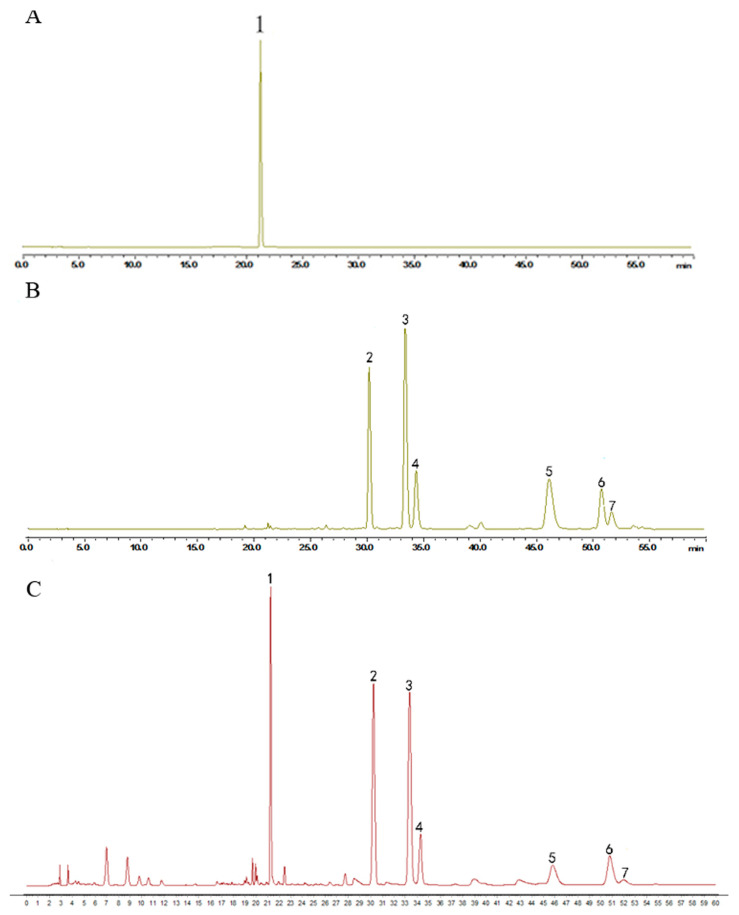
HPLC chromatogram of (**A**) xantholhumol; (**B**) the mixed standard control and (**C**) the hops extract. The content of xanthohumol in the extract was 0.55%. (1. xanthohumol; 2. cohumulone; 3. humulone; 4. adhumulone; 5. colupulone; 6. lupulone; 7. adlupulone).

## Data Availability

The data presented in this study are available on request from the corresponding author.

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
