# Peer review of "Humulus lupulus L. Extract Protects against Senior Osteoporosis through Inhibiting Amyloid β Deposition and Oxidative Stress in APP/PS1 Mutated Transgenic Mice and Osteoblasts"

_molecules, 2023, doi:10.3390/molecules28020583_

Round 1

Reviewer 1 Report

This manuscript firstly  demonstrate that hops extract protects against Aβ-induced senior osteoporosis primarily by ameliorating memory impairment, enhancing BMD and trabecular bone structure in APP/PS1 mutated transgenic mice. The manuscript looks good but still need further improvement.

1.       Line 37 Alzheimer’s disease (AD), while line 58 Alzheimer’s disease.

2.       Figure 1a. It’s hard to distinguish each treatment. Please use different color to represent different treatment.

3.       Results 2.1. please simply describe how these treatments were performed. It may not as be detailed as that in methods. at least it should include this information: How old the mice are when began treatment, how to do treatment? How long? Line 85. Please explain what NAC is. And why include NAC treatment? As positive control? Or negative control.

4.       Figure 2. I can’t see clear Aβ plaque staining in mice brain. It may be due to the low quality of the picture or staining. In addition, it is impossible to find Aβ plaque staining in wild type mice. Thus, the staining result looks have some unspecific staining. please perform brain IHC or Thioflavin S staining instead.

5.       Please include full name on first occurrence of BVF, BMD, SOD, OCN as well as other abbreviations.

6.       Line 157, 161 162. “hops” should be “HLE”

7.       Figure 6. Please describe what “model” group is.

8.       Line 290. 10 mice for each group, but why in figure 1, 2,3 and 4 n=8, 3, 8 and 3 respectively?

9.       Line 319 “three sections for one mouth” should be “three sections for one mouse” and is this the reason why n=3 in Figure 2 and 4?

Author Response

Thank you for the reviewer’s useful comments and suggestions concerning our manuscript entitled “Humulus lupulus L. extract protects against senior osteoporosis through inhibiting amyloid β deposition and oxidative stress in APP/PS1 mutated transgenic mice and osteoblasts” (molecules-2143678). Those comments are all valuable and helpful for revising and improving our paper, as well as the important guiding significance to our researches. We have modified the manuscript accordingly, and the revised portion were marked using the “Track Changes” in the revised manuscript. The detailed corrections are listed below point by point:

Reviewers' comments:

Reviewer: 1

Comments to the Author

This manuscript firstly  demonstrate that hops extract protects against Aβ-induced senior osteoporosis primarily by ameliorating memory impairment, enhancing BMD and trabecular bone structure in APP/PS1 mutated transgenic mice. The manuscript looks good but still need further improvement.

  1. Line 37 Alzheimer’s disease (AD), while line 58 Alzheimer’s disease.

Response: Thanks for your suggestion. We have corrected this error in the revised manuscript.

  1. Figure 1a. It’s hard to distinguish each treatment. Please use different color to represent different treatment.

Response: Thanks for your suggestion. We have corrected Figure 1a in the revised manuscript.

  1. Results 2.1. please simply describe how these treatments were performed. It may not as be detailed as that in methods. at least it should include this information: How old the mice are when began treatment, how to do treatment? How long? Line 85. Please explain what NAC is. And why include NAC treatment? As positive control? Or negative control.

Response: Thanks for your suggestion. We have described how Morris water maze task performed in Results 2.1 in the revised manuscript. In addition, NAC (N-Acetyl-L-cysteine) is a strong antioxidant, which was used as a positive control drug in this study. We have explained it in 2.1 in the revised manuscript.

  1. Figure 2. I can’t see clear Aβ plaque staining in mice brain. It may be due to the low quality of the picture or staining. In addition, it is impossible to find Aβ plaque staining in wild type mice. Thus, the staining result looks have some unspecific staining. please perform brain IHC or Thioflavin S staining instead.

Response: Thanks for your suggestion. In figure 2a, the brownish red or orange precipitates were positive for Aβ. Maybe the picture is a little small, and you can enlarge the picture. There indeed some red dots in hippocampus and cortex of mice, especially in APP/PS1 group. In addition, Aβ plaque in wild mice may be a rare phenomenon due to the aging of mice, which were also discovered in other previous studies (doi:10.1016/j.phymed.2021.153780, doi:10.1016/j.intimp.2021.108002, doi:10.1186/s12974-020-01981-4). About the brain IHC or Thioflavin S staining, it is really a very good suggestion, and we will perform it in the following studies.

  1. Please include full name on first occurrence of BVF, BMD, SOD, OCN as well as other abbreviations.

Response: Thanks for your suggestion. We have supplemented these abbreviations in the revised manuscript.

  1. Line 157, 161 162. “hops” should be “HLE”

Response: Thanks for your suggestion. We have corrected this error in the revised manuscript.

  1. Figure 6. Please describe what “model” group is.

Response: Thanks for your suggestion. We have described it in the figure caption of figure 6 in the revised manuscript.

  1. Line 290. 10 mice for each group, but why in figure 1, 2,3 and 4 n=8, 3, 8 and 3 respectively?

Response: Thanks for your suggestion. At the beginning, we designed 10 mice for each group. With the progress of the administration, some mice died, and there are at least 8 mice in each group in the end. In order to ensure the same number of mice in each group, 8 mice were selected for in vivo experiments. In addition, as for Congo red strain and immunohistochemistry experiments, the minimum number of samples required is 3. Considering that other relevant experiments may be carried out in the future, 3 samples were selected for Congo red strain and immunohistochemistry experiments.

  1. Line 319 “three sections for one mouth” should be “three sections for one mouse” and is this the reason why n=3 in Figure 2 and 4?

Response: Thanks for your suggestion. We have corrected this error in the revised manuscript. As for the reason why n=3 in Congo red strain and immunohistochemistry experiments, we have described it in the last question. Other samples were reserved for the following experiments.

Reviewer 2 Report

The manuscript reports some interesting results that it was the first time to demonstrate that hops extract protects against Aβ-induced senior osteoporosis primarily by ameliorating memory impairment, enhancing BMD and trabecular bone structure in APP/PS1 mutated transgenic mice, and improving osteoblastic MC3T3-E1 cells activities. As for its mechanism, removing Aβ deposition in brain and bone and its inducing oxidative stress plays a crucial role.

In general, this paper is clearly laid out, well planned and easy to read. The experiments are well designed and appropriate. Some specific and general suggestion are listed below:

1.      In the introduction section, line 33, the author mentioned that ageing is a major pathogenic factor that causes SOP. The author should provide recent studies with references and medical evidence.

2.      In paragraph line 69, the author stated that the antioxidant properties of the examined plant were used to check memory, bone deficiency, and Aβ absorption. The author should specify their hypothesis with recent research relating to that disease and provide references.

  1. There are a few typo mistakes. So, the author should thoroughly read the entire paper and correct typographical errors.
  2. There are few grammatical mistakes. So, the author should correct grammatical errors. Accepted for publication. 

Author Response

Thank you for the reviewer’s useful comments and suggestions concerning our manuscript entitled “Humulus lupulus L. extract protects against senior osteoporosis through inhibiting amyloid β deposition and oxidative stress in APP/PS1 mutated transgenic mice and osteoblasts” (molecules-2143678). Those comments are all valuable and helpful for revising and improving our paper, as well as the important guiding significance to our researches. We have modified the manuscript accordingly, and the revised portion were marked using the “Track Changes” in the revised manuscript. The detailed corrections are listed below point by point:

Reviewer: 2

Comments to the Author

The manuscript reports some interesting results that it was the first time to demonstrate that hops extract protects against Aβ-induced senior osteoporosis primarily by ameliorating memory impairment, enhancing BMD and trabecular bone structure in APP/PS1 mutated transgenic mice, and improving osteoblastic MC3T3-E1 cells activities. As for its mechanism, removing Aβ deposition in brain and bone and its inducing oxidative stress plays a crucial role.

In general, this paper is clearly laid out, well planned and easy to read. The experiments are well designed and appropriate. Some specific and general suggestion are listed below:

  1. In the introduction section, line 33, the author mentioned that ageing is a major pathogenic factor that causes SOP. The author should provide recent studies with references and medical evidence.

Response: Thanks for your suggestion. We have supplemented relevant contents in the introduction section of revised manuscript.

  1. In paragraph line 69, the author stated that the antioxidant properties of the examined plant were used to check memory, bone deficiency, and Aβ absorption. The author should specify their hypothesis with recent research relating to that disease and provide references.

Response: Thanks for your suggestion. We have introduced it in the second paragraph of introduction section, and supplemented some content according to your suggestion.

  1.  There are a few typo mistakes. So, the author should thoroughly read the entire paper and correct typographical errors.

Response: Thanks for your suggestion. We have thoroughly read the entire paper and corrected some typographical errors.

  1.  There are few grammatical mistakes. So, the author should correct grammatical errors. Accepted for publication. 

Response: Thanks for your suggestion. We have thoroughly read the entire paper and corrected some grammatical errors.

Round 2

Reviewer 1 Report

I still have concern about Figure2A, authors provide 3 references to argue that WT mice also have amyloid plaque staining. However, the first reference (doi:10.1016/j.phymed.2021.153780) the results quality is too low to believe. while, for the last two references, even though they show some amyloid plaque signal in WT mice,  it's much much lower than APP/PS1 mice. After, discuss with my colleague, we still believe that old WT mice only have very rear or no amyloid plaque. So I strongly suggest that the authors use the other staining methods to further confirm the results!!!

Author Response

Thank you very much for your comment. In fact, we also think that WT mice only have rear amyloid plaque. During the analysis of experimental data, we found a few amyloid plaque in old WT mice, however, this number is very small compared with that in APP/PS1 mice. In addition, this paper mainly focuses on the effect of hops on protecting against osteoporosis, and Aβ deposition is only one of factors leading to osteoporosis. So we think it is an acceptable result in this paper. Certainly, we strongly agree with you that other staining methods should be used, however, due to the time limit for revision, the relevant results may be difficult to be presented in this paper. Recently, we are designing some experiments to deeply study the relationship between Alzheimer's disease and osteoporosis, and some staining methods, such as IHC and Thioflavin S staining will be performed in the following study. I also hope that I will have opportunity to discuss with you and learn from you in the future.